# Counterfactual analysis of differential comorbidity risk factors in Alzheimer's disease and related dementias

**Yejin Kim**[1,2]*, **Kai Zhang**[1], **Sean I. Savitz**[2], **Luyao Chen**[1], **Paul E. Schulz**[3], **Xiaoqian Jiang**[1,2]

**1** School of Biomedical Informatics, University of Texas Health Science Center at Houston, Houston, Texas, United States of America, **2** Institute for Stroke and Cerebrovascular Disease, University of Texas Health Science Center at Houston, Houston, Texas, United States of America, **3** Department of Neurology, University of Texas Health Science Center at Houston, Houston, Texas, United States of America

* Yejin.Kim@uth.tmc.edu

**Data Availability Statement:** We used Cerner Health Facts, a clinical database covering electronic health records (EHRs) from Cerner client hospitals. The Cerner Health Facts database contains a de-

## Abstract

Alzheimer's disease and related dementias (ADRD) is a multifactorial disease that involves several different etiologic mechanisms with various comorbidities. There is also significant heterogeneity in the prevalence of ADRD across diverse demographics groups. Association studies on such heterogeneous comorbidity risk factors are limited in their ability to determine causation. We aim to compare counterfactual treatment effects of various comorbidity in ADRD in different racial groups (African Americans and Caucasians). We used 138,026 ADRD and 1:1 matched older adults without ADRD from nationwide electronic health records, which extensively cover a large population's long medical history in breadth. We matched African Americans and Caucasians based on age, sex, and high-risk comorbidities (hypertension, diabetes, obesity, vascular disease, heart disease, and head injury) to build two comparable cohorts. We derived a Bayesian network of 100 comorbidities and selected comorbidities with potential causal effect to ADRD. We estimated the average treatment effect (ATE) of the selected comorbidities on ADRD using inverse probability of treatment weighting. Late effects of cerebrovascular disease significantly predisposed older African Americans (ATE = 0.2715) to ADRD, but not in the Caucasian counterparts; depression significantly predisposed older Caucasian counterparts (ATE = 0.1560) to ADRD, but not in the African Americans. Our extensive counterfactual analysis using a nationwide EHR discovered different comorbidities that predispose older African Americans to ADRD compared to Caucasian counterparts. Despite the noisy and incomplete nature of the real-world data, the counterfactual analysis on the comorbidity risk factors can be a valuable tool to support the risk factor exposure studies.

## Author summary

Alzheimer's disease (AD) is the sixth leading cause of death in the United States, affecting 6 million Americans aged 65 and older. AD risk develops over the long course of a lifetime

identified EHR and is subscribed by the University of Texas Health Science Center for research use. To apply, contact https://www.cerner.com/ap/en/solutions/data-research.

**Funding:** YK is supported in part by UTHealth startup, UT Stars award, and the National Institute of Health (NIH) under award number R01AG066749. SIS is supported by the Frank M. Yatsu Chair in Neurology. PES is supported by the Weston Brain Institute, The Kleberg Foundation, donations, and numerous pharmaceutical companies. XJ is CPRIT Scholar in Cancer Research (RR180012), and he was supported in part by Christopher Sarofim Family Professorship, UT Stars award, UTHealth startup, the National Institute of Health (NIH) under award number R01AG066749 and U01TR002062. The funders had no role in the design, methods, subject recruitments, data collections, analysis, and preparation of the paper.

**Competing interests:** The authors have declared that no competing interests exist.

and involves various etiologies such as genetic, vascular, and psychosocial factors, of which the complex biological mechanisms are still under investigation. Putative risk factors include race/ethnicity, low educational attainment, socioeconomic status, and comorbidities (hypertension, diabetes). These risk factors may interact with each other and further increase the risk of AD. Most studies find older African Americans are more likely than older Whites to develop AD. Comorbidity risk factors and socioeconomic status are believed to partially account for these differences, as they are more prevalent in African Americans. To disentangle the multifactorial effects of factors predisposing older adults to AD, we quantified counterfactual effect of high-risk comorbidities mediating the AD risk using nationwide electronic health records. We particularly focused on differential counterfactual effects between matched African Americans and Caucasians. Our extensive counterfactual analysis discovered different comorbidities that predispose older African Americans to AD compared to Caucasian counterparts. This differential risk between racial groups will contribute to developing targeted treatment to AD.

## Introduction

Alzheimer's disease (AD) is the 6th leading cause of death in the United States and it is the only one of the top 10 leading causes of death that cannot be cured [1–3]. Alzheimer's disease and related dementias (ADRD) is a *multifactorial* disease that involves several different etiologic mechanisms with highly heterogeneous phenotypes [4,5]. Moreover, prior studies suggest that there is significant *heterogeneity* in the prevalence of ADRD across diverse demographic groups [6,7]. For example, most studies find that older African Americans are more likely than older non-Hispanic Caucasians to be diagnosed with ADRD [7–10]. Comorbidity risk factors such as cardiovascular disease, diabetes, and obesity, as well as socioeconomic status, are believed to account for these differences, as they are more prevalent in African Americans [1,2].

Association studies on such multifactorial and heterogeneous comorbidity risk factors are limited in their ability to determine causation. For example, although obesity is associated with increasing ADRD risk [11], its effect may be mediated by comorbidities such as hypertension, cardiovascular disease, and diabetes [12–14]. Counterfactual analysis, on the other hand, uses a methodology to estimate the outcome for an individual who had been exposed to a risk factor (factual) under alternative exposure scenarios (counterfactual) of if the individual had not been exposed. A confounder is a variable causing exposure to the risk factors and also outcomes. It is a major source of bias that can mislead us to draw wrong conclusions that the risk factor causes the outcome when it does not [15]. The gold standard to avoid such bias is to randomize the exposure in randomized clinical trials, but such a randomized study is not feasible in studying risk factors, particularly when the exposure is unethical (e.g., exposing subjects to putative risk factors) [16,17]. Alternatively, the counterfactual analysis with observational data aims to reduce the bias by adjusting the distribution of conditions that affect the exposure to the risk factor, such as via propensity score matching or weighting [18].

The statistical inferences to estimate the causality and counterfactuals from observational data in medicine have been long discussed but not yet widely used [16,18–25]. For example, previous research proposes a framework for emulating randomized trials from big observational data [17,18,26]. This framework simulates randomized clinical trials via controlling baseline characteristics, identifying time zero (baseline) to the outcome, adjusting the confounders by matching, and estimating treatment effect using potential outcome models [17]. A

challenge here is that latent confounders can lead to selection bias; the causal structure can delineate the relationship between these confounders and help reduce the selection bias [27]. Indeed there is a separate line of causal analysis studies utilizing causal structure learning to investigate conditional independence among comorbid conditions [28–33], mainly with a few predetermined selected variables due to super-exponential complexity in structure learning. To date, comorbidity risk factor studies in ADRD often focus on the association [1,2], rather than causation [24,34]. Similar to the emulation of randomized clinical trials [17], the goal of this study was to investigate the counterfactual effect of comorbidity risk factors in ADRD, particularly focusing on racial heterogeneity (Fig 1).

One challenge to counterfactual comorbidity risk factor studies is the lack of data capturing comprehensive health conditions before ADRD onset. As ADRD is a heterogeneous disease with various etiologies, counterfactual analysis requires an extensive set of comorbidities to investigate how one disease might contribute to ADRD. Voluminous electronic health records (EHRs) from nationwide hospitals are a rich source for providing comprehensive data on the risk of ADRD. Nationwide EHRs also have larger sample sizes even in the minority populations compared to the sample size in data in clinical trials or observational studies, in which a participation rate is significantly low in the minority populations [35,36]. EHR data, however, are mainly collected for billing, not for scholarly study, and thus diagnosis billing codes in EHRs are sometimes incomplete and lack important details such as socioeconomic factors (e.g., education, literacy, life course exposures), which are one of the main causes of racial disparities in ADRD [37,38]. Despite the potential limitation due to these unobserved confounders, EHRs can extensively cover a large population's long medical history in breadth and provide us a unique opportunity to investigate the counterfactual effect of comorbidity risk factors for ADRD. We undertook this study to provide unbiased insights on the racial differences of comorbidity risk factors in ADRD.

## Materials and study design

### Database

We utilized Cerner Health Facts, a large clinical database covering EHRs from more than 600 Cerner client hospitals, from 2000–2017, with a total of 49,826,000 inpatients and outpatients (Fig 1A) [39]. The Cerner Health Facts database contains a de-identified EHR and is subscribed by the University of Texas Health Science Center for research use [39]. These nationwide multi-center EHRs can increase generalizability of our findings.

### Observation period

We summarized our study design in Table 1. We included subjects with observation after the age of 65 and observations longer than 6 months. Age is the strongest risk factor for ADRD. Non-ADRD subjects were either not old enough to have ADRD onset (e.g., average ADRD onset age was 79.99 for African Americans and 81.57 for Caucasians) or old enough but censored (e.g., median observation length was 1.0 years). To avoid bias caused by different age distributions in ADRD and non-ADRD subjects, we matched age in their observation period (Fig 1B, 1C Matching 1). That is, for the ADRD subjects, the observation window started from when any diagnosis code was first recorded and ended when the first ADRD onset was recorded (Fig 1D). For the non-ADRD subjects, we selected subjects that had the closest age at the observation starts and ends. We truncated non-ADRD observations after the age when matched ADRD observations ended (Fig 1D).

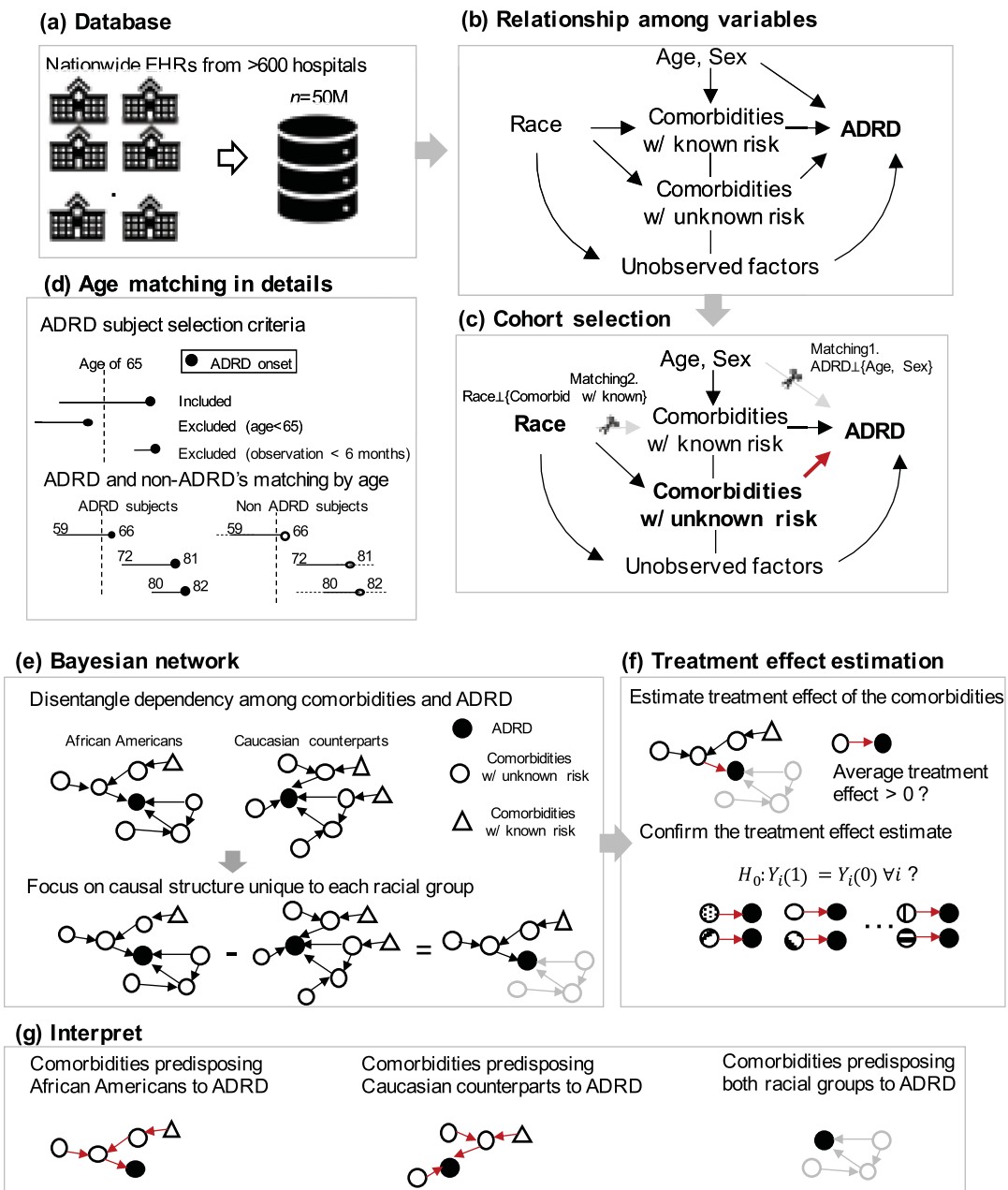

**Fig 1. Study overview.** Our goal is to assess the counterfactual effect of comorbidities that predispose each racial group to ADRD. We focused on African Americans and non-Hispanic Caucasians. (a) We used Cerner EHRs from more than 600 Cerner client hospitals. (b,c) To select cohort, we matched age and sex in ADRD and non-ADRD subjects. To make African American and Caucasian cohorts comparable, we matched race on the known ADRD risk factors (age, sex, hypertension, diabetes, obesity, heart disease, vascular disease, and head injury). Hidden confounders (such as socioeconomic status) were presented for clarification. (d) Age is the strongest risk factor in ADRD. We matched the age of ADRD and non-ADRD subjects. (e) To disentangle the multifactorial effect of comorbidities, we derived a Bayesian network of comorbidities and ADRD using constraint-based algorithms. (f) We used inverse probability of treatment weighting to estimate the counterfactual effect of comorbidities that have a direct edge to ADRD in each racial group's Bayesian network. We performed the permutation test to validate the counterfactual effect. (g) We examined the difference in comorbidities paths of the Bayesian network.

**Table 1. Summary of the study design based on Target Trial framework [17].** We performed the analysis for African American and Caucasian counterparts, respectively.

| Protocol component | Description |
|---|---|
| Eligibility criteria | Older adults with the age of 65 and older during the observation between 2000–2017. Observation length >6 months |
| Treatment strategies | Exposure to comorbidity. The comorbidities are selected based on the structural causal model (i.e.., Bayesian network) |
| Assignment procedures | To emulate random assignment of treatment, confounding variables were adjusted via propensity score matching, inverse weighting, or stratification. Confounding variables were identified based on structural causal model (i.e., Bayesian network) |
| Sampling strategies | 100 bootstraps by random selection with replacement. 95% confidence interval reported |
| Follow-up period | Follow-up ends at ADRD onset (for ADRD subjects) and similar ADRD onset age (for non-ADRD subjects). Follow-up starts when the first comorbidities are recorded. The median follow-up period was 1.0 years. |
| Outcome | ADRD onset after the comorbidities. ADRD onset identified by multiple PheWas codes and medication codes |
| Analysis plan | Compare the average number of ADRD onset between the comorbidity exposure group and non-exposure group |

## Exposure to comorbidities

Comorbidities of interest were all the diseases (identified as diagnosis codes) that were diagnosed within the observation period, which might have potential risk to predispose to ADRD onset. EHRs are inherently noisy and sparse. To accurately identify the most direct putative risk factors and better understand root causes, it is important to condense the sparse diagnosis codes into clinically meaningful comorbidities and disentangle the effects of multifactorial comorbidities. Our approach to addressing this challenge was to group ICD9 or ICD 10 diagnosis codes by PheWas hierarchy to increase clinical relevance of the billing codes [40]. PheWas code is a hierarchical grouping of ICD codes based on statistical co-occurrence, code frequency, and human review. For more detail, see reference [40]. We included 100 PheWas disease codes that appeared within the observation window in more than 5% of the subjects. We counted the occurrence of each disease code that appears during the observation and converted them to a logarithm scale (i.e., log2(1+counts)) as the count distributions are skewed.

## Outcome

The outcome of interest was ADRD onset, which we detected as having either ADRD diagnosis code or medication. The ADRD diagnosis codes were PheWas codes for 290.11 (Alzheimer's disease); 290.12 (Frontotemporal dementia, Pick's disease, Senile degeneration of brain); 290.13 (Senile dementia); and 290.16 (Vascular dementia, Vascular dementia with delirium/delusions/depressed mood). The ADRD medications were acetylcholinesterase inhibitors (Donepezil, Galantamine, and Rivastigmine) or memantine. The definition of ADRD in EHRs can be controversial considering the fact that EHRs are for billing purposes. We compared our definition of ADRD onset with other potential definitions and discussed our rationale behind choosing our definition considering racial bias in S1 Text A.

## Methods

We investigated and compared differences in the comorbidities predisposing each racial group to ADRD. Our approach for counterfactual effects analysis has three steps: i) cohort selection by matching sex, age, and known comorbidity risk factors, ii) disentangle dependency among

comorbidities and ADRD to identify core skeleton that accounts for increasing ADRD risk by Bayesian network, and iii) measure and validate the counterfactual effects of the identified comorbidities with a direct relationship to ADRD by inverse probability of treatment weighting. Code is publicly available at https://github.com/yejinjkim/treatment-effect.

## Matching to obtain comparable racial cohorts

We matched African Americans and non-Hispanic Caucasians in terms of age, sex, and known comorbidity risk factors to create comparable racial cohorts (Fig 1C). Direct comparison on the risk of ADRD between African Americans and Caucasians would produce biases due to confounders in the large-scale heterogeneous EHRs [41,42]. For example, in Cerner Health Facts EHRs, 68.1% were female in African Americans, whereas 62.8% were female in Caucasians (Table 2). Careful cohort matching is needed to build comparable cohorts that are similar in terms of known risk factors. Theoretically, this cohort matching is not necessary for the Bayesian network (discussed in the next section) because the Bayesian network captures local interaction between variables, but we found that this cohort matching is helpful to the unconfoundedness assumption [16].

We first matched ADRD subjects and non-ADRD subjects based on age and sex for each racial group. The comorbidities (with either known or unknown risk) incidence differs by race. Our focus is to identify the effect of comorbidities with an unknown risk that disproportionately affects racial groups because we already know the differential effect of comorbidities with known risk. The comorbidities that are known to increase ADRD risk disproportionately among racial groups were hypertension, diabetes, obesity, heart disease, vascular disease, and head injury (specific definition of each comorbidity is in S1 Table) [1,2]. So, we matched African Americans and Caucasians based on the comorbidities with known risk using propensity scores (Matching 2 in Fig 1C), so that the remaining potential risk factors are isolated. We also matched African Americans and Caucasians based on age and sex. We used the nearest

**Table 2. Cohort demographics.** Distribution of comorbidities with a known risk before and after matching between African Americans and Caucasians. Standardized bias = difference in the mean of a given variable between African Americans and Caucasians divided by the standard deviation in African Americans.

| | Before | | | | Stand. bias | After | | | | Stand. bias |
|---|---|---|---|---|---|---|---|---|---|---|
| | African American | | Caucasian | | | African American | | Caucasian | | |
| | ADRD | Non ADRD | ADRD | Non ADRD | | ADRD | Non ADRD | ADRD | Non ADRD | |
| Total subjects | 16,215 | 14,940 | 111,690 | 105,931 | | 7,662 | 7,418 | 7,869 | 7,357 | |
| Mean age at observation end (std) | 79.77 (6.96) | 82.87 (7.12) | 81.57 (6.54) | 80.81 (6.47) | 0.0553 | 80.55 (7.02) | 83.33 (7.11) | 81.59 (6.52) | 82.80 (6.81) | 0.0432 |
| Female<br>Male | 10,953 (67.6%)<br>5,261 (32.4%) | 10,249 (68.6%)<br>4,691 (31.4%) | 70,217 (62.9%)<br>41,473 (37.1%) | 66,552 (62.8%)<br>39,378 (37.2%) | 0.1097 | 5,249 (68.5%)<br>2,413 (31.5%) | 5,169 (69.7%)<br>2,249 (30.3%) | 5,438 (69.1%)<br>2,431 (30.9%) | 5,127 (69.7%)<br>2,230 (30.3%) | 0.0018 |
| Hypertension | 8,828 (54.4%) | 5,041 (33.7%) | 57,410 (51.4%) | 45,335 (42.8%) | 0.0541 | 5,315 (69.4%) | 3,207 (43.2%) | 4,984 (63.3%) | 3,346 (45.5%) | 0.0273 |
| Diabetes | 5,380 (33.2%) | 2,776 (18.6%) | 24,758 (22.2%) | 17,959 (17.0%) | 0.1563 | 2,725 (35.6%) | 1,481 (20.0%) | 2,413 (30.7%) | 1,474 (20.0%) | 0.0409 |
| Heart disease | 4,420 (27.3%) | 2,207 (14.8%) | 33,037 (29.6%) | 24,438 (23.1%) | 0.1208 | 2,551 (33.3%) | 1,301 (17.5%) | 2,192 (27.9%) | 1,301 (17.7%) | 0.0408 |
| Vascular disease | 4,349 (26.8%) | 1,819 (12.2%) | 31,978 (28.6%) | 21,266 (20.1%) | 0.1126 | 2,485 (32.4%) | 1,054 (14.2%) | 2,175 (27.6%) | 1,100 (15.0%) | 0.0354 |
| Obesity | 880 (5.4%) | 671 (4.5%) | 4,125 (3.7%) | 3,835 (3.6%) | 0.065 | 341 (4.5%) | 264 (3.6%) | 237 (3.0%) | 203 (2.8%) | 0.0516 |
| Head injury | 474 (2.9%) | 218 (1.5%) | 5,010 (4.5%) | 2,018 (1.9%) | 0.062 | 260 (3.4%) | 108 (1.5%) | 219 (2.8%) | 75 (1.0%) | 0.0256 |

neighbor matching with radius and caliper [43]. We reported standardized bias to evaluate the balance between the two groups. Detailed cohort selection process with the structural equation is available in S1 Text B.

## Identify comorbidities with potential causal effect

Using the matched cohorts of African Americans and non-Hispanic Caucasians, we aimed to identify comorbidities that predispose each racial group to ADRD with potential causal effects. We derived the Bayesian network (Fig 1E), a directed acyclic graph of comorbidities and ADRD that has directed edges implying causation. Learning Bayesian networks is a principled approach to identifying and analyzing multifactorial effects, as it takes other confounders into account to determine possible causal effects via considering conditional independence. We built two Bayesian networks for African Americans and Caucasian counterparts respectively. Nodes were all comorbidities and ADRD. We set three tiers: *tier1* = comorbidities with known risk, *tier2* = comorbidities with unknown risk, and *tier3* = ADRD. The comorbidities in tier1 were mostly chronic diseases that do not have a direct or immediate effect on ADRD (e.g., hypertension, diabetes) but have an indirect effect on ADRD by mediating through other subsequent comorbidities. The PC algorithm is one of the principled causal structure learning algorithms (by Peter and Clark) that can be applied to find Bayesian Networks (details in S1 Text C). [44,45] PC has been implemented by various software/libraries such as TETRAD [46], *pcalg*,[46,47] *bnlearn*,[48] and speed-up versions by Zarebavani and Zhang [49,50].

The causal structure, however, can vary by input data, which hinders the robustness of the structure. Bootstrapping and graph combination methods are usually adopted to increase robustness of the inference results [51,52,53]. We used the voting-based causal graph combination to obtain a robust and unbiased estimation of the causal graph, which significantly reduce the false positives and increase the overall robustness of the graph learning [54]. We leveraged the majority voting technique by randomly splitting the entire data into ten sub-datasets while withholding 10% of the original data each time. We applied the PC algorithm (with a significance level of 0.05) on each of the ten sub-datasets and aggregated the ten results. A directed edge presented in the final ensembled causal graph if it appears in more than half of all the causal graphs. We repeated the same procedure on each racial group and derived the final causal graphs of the two racial groups.

After we obtained the robust causal graphs, we investigated whether the two causal graphs are distinct enough. To compare the difference of the causal graphs, we used two metrics: structural hamming distance (SHD) and the graph edit distance (GED). The SHD measures the number of edges in which the two compared graphs do not coincide. The GED measures length of the shortest graph edit path, which is a sequence of node and edge edit operations (including substitutions, deletions, and insertions) transforming graph G1 to graph isomorphic to G2 [55].

## Treatment effects of identified comorbidities on ADRD risk

After we identified unique comorbidities that predispose older African Americans and Caucasian counterparts to ADRD respectively, we quantified the causal effect of the identified comorbidities on ADRD risk by measuring the average treatment effect on increasing (or decreasing) ADRD risk (Fig 1F). That is, we would like to answer this question: "If a subject who had been exposed to the comorbidity in fact did not have the comorbidity (counterfactual), would the subject have had a lower level of ADRD risk?" The treatment effect analysis is to identify the difference in potential outcomes (e.g., ADRD risk) when the subject is exposed and not exposed to the comorbidity. Let us denote $Y_i(1)$ subject *i*'s outcome (i.e., ADRD onset)

when the subject has certain comorbidity ($T_i = 1$) and $Y_i(0)$ subject $i$'s outcome when the subject does not have the comorbidity ($T_i = 0$). The treatment effect $\tau_i$ of having this comorbidity is then defined as: $\tau_i = Y_i(1) - Y_i(0)$ based on Neyman-Rubin's potential outcome models [56]. Obviously, it is impossible to observe factual and counterfactual outcomes at the same time (e.g., it is impossible that a subject does and does not have the comorbidity, $Y_i(1)$ and $Y_i(0)$). An approach to mitigate this missing counterfactual outcome is to average out the potential outcomes in the exposed and the unexposed respectively and estimate the average treatment (ATE) effect by $E[Y(1)] - E[Y(0)]$. The ATE is an unbiased estimate of the treatment effect (i.e., effect of comorbidity exposure) if the subjects are randomly assigned to either exposure or control group (such as in randomized clinical trials). However, in real-world data the exposure to the comorbidity is not random; subjects with and without the comorbidity differ systematically. The way to reduce the bias between the two groups is to match the subjects or weight their outcomes $Y$ based on the likelihood of having the comorbidity $T$ so that the likelihood distributions are similar [19,20]. Here we denote the likelihood of having the comorbidity $T$ given subject's condition $X$ as propensity score. Several strategies to estimate the unbiased treatment effect include propensity score matching (to directly obtain counterfactual outcome by identifying propensity-matched neighbors), propensity score stratification (to stratify subjects into groups with a similar level of propensity score and directly compare the outcomes within each group), and inverse probability of treatment weighting (IPTW) [19,20]. Specifically we used the IPTW [57], which down-weights over-sampled patients and up-weights under-sampled subjects so that the two groups with and without the comorbidity are similar. The ATE can be then estimated as

$$\text{ATE} = 1/n \sum_{i=1}^{n} [T_i Y_i / e(X_i) - (1 - T_i) Y_i / (1 - e(X_i))],$$

where $e(X_i)$ is the propensity scores at $T_i = 1$ given subject's features $X_i$. We are more interested in treatment effect among those who already had the comorbidity $T_i = 1$, which is the so-called average treatment effect among treated (ATT). We can estimate ATT by:

$$\text{ATT} = 1/n_1 \sum_{i=1}^{n_1} [T_i Y_i - e(X_i)(1 - T_i) Y_i / (1 - e(X_i))],$$

where $n_1$ refers to the number of subjects with $T_i = 1$. We can similarly define the average treatment effect among untreated or control (ATC) among those who did not have the comorbidity $T_i = 0$. In all, the treatment effect measures the amount of difference in outcome $Y$ due to exposure $T$ given the similarly weighted conditions $X$. For example, ATE = $\tau > 0$ means that the outcome when subjects are intervened to have the comorbidity is greater by $\tau$ than the outcomes when subjects are intervened not to have the comorbidity, implying the comorbidity exposure increases the outcomes on average. Similarly, ATT = $\tau > 0$ means that the outcome of subjects already having the comorbidity is greater by $\tau$ than the outcomes when the subjects are intervened not to have the comorbidity, implying the subjects with the comorbidity would have decreased the outcome (ADRD onset) by $\tau$ if they are without the comorbidity.

We obtained the propensity score $e(X_i)$ of having the comorbidity $T$ given subject's features $X$ using logistic regression. To avoid high variance of propensity scores due to overfitting, we used the self-normalized propensity estimator (or Hàjek estimator) [58]. The subject's features $X_i$ to infer propensity scores were the all other remaining comorbidities that have directed edge to $T$ (the comorbidity of interest) and $Y$ (ADRD) in the derived Bayesian network. Here we can also measure ATE and ATT of all the pairs of comorbidities with the directed edges in the Bayesian network to quantify the causal effect of one comorbidity to the others. We calculated the 95% confidence interval of the ATE and ATT with 100 bootstraps by random

selection. We measure the ATE and ATT for each racial group. We used *Dowhy*, a publicly available package to measure the treatment effects [59].

**Confirm the estimated treatment effect via the permutation test.** We confirmed the estimated treatment effect via the permutation test (Fig 1F) [60,61]. The permutation test (or randomization test) is to assess if an ATE estimate is statistically significant by testing for Fisher's Sharp Null, $H_0$: $Y_i(1) = Y_i(0)$, $\forall i$, which states that there is no treatment effect for all subjects [62]. A rejection of Fisher's Sharp Null means there is a significant treatment effect [61]. We randomly shuffled the treatment variable (binary indicator whether the subject has the comorbidity or not) to make the treatment variable independent and observed the treatment effect as repeating the random permutation. We set 100 repetitions and used *Dowhy* to implement the permutation tests [59].

## Results

### Study subjects

We first built matched cohorts of ADRD and non-ADRD subjects. Of the 49,826,000 patients, there were 157,620 subjects with ADRD diagnosis codes; 163,320 subjects with ADRD medication codes; and 235,912 subjects with either the diagnosis or medication codes. After excluding subjects without diagnosis/medication codes, timestamp, and observation length less than 6 months, we selected 138,026 ADRD subjects and matched 138,026 non-ADRD subjects based on age and sex.

### Cohort identification

We then matched African American and non-Hispanic Caucasian groups to build comparable cohorts of the two racial groups. After the extensive matching and reducing confounding effects, the final cohort was 7,662 ADRD and 7,418 non-ADRD for African Americans; 7,869 ADRD and 7,357 non-ADRD for non-Hispanic Caucasians. We calculated the standardized bias of each variable between African Americans and Caucasians to check whether the variables are balanced. The standardized bias < 0.10 was used as the cutoff value to confirm the balance after matching [63]. As a result, the age and comorbidity distributions were similar between the racial groups (Table 2, Fig 2); Most variables had standardized bias <0.10 after matching.

### Bayesian Network and counterfactual effect of comorbidities

We derived the Bayesian network of comorbidities and ADRD for each racial group separately (Fig 3, S2 Table). The two causal structures from African Americans and their matched Caucasian cohort shared similar but also distinctive comorbidities. The SHD and GED of the two structures was 1,277 and 895, respectively. Among all the edges in the Caucasian causal graph, only 46.81% edges are present in the African American causal graph; and only 36.59% of the edges in the African American causal graph are present in the Caucasian causal graph. We focused on the comorbidities that have edges to ADRD in all ten bootstraps (Table 3) and measured the counterfactual treatment effect of the comorbidities to ADRD (Table 4, Fig 3). The identified comorbidities were grouped into three types: cerebrovascular disease, mental disorders, and inflammation/infection.

**Cerebrovascular disease (CVD).** In contrast to Caucasians, African Americans had an edge from *the late effects of CVD (*PheWas Code = *433.8*, specific diseases it covers are in S3 Table) to ADRD. Its ATE (by IPTW [64]) was 0.2715 (Table 4), which implies that having this comorbidity increases the risk for ADRD by 27.15% point on average. The ATT of 0.1908

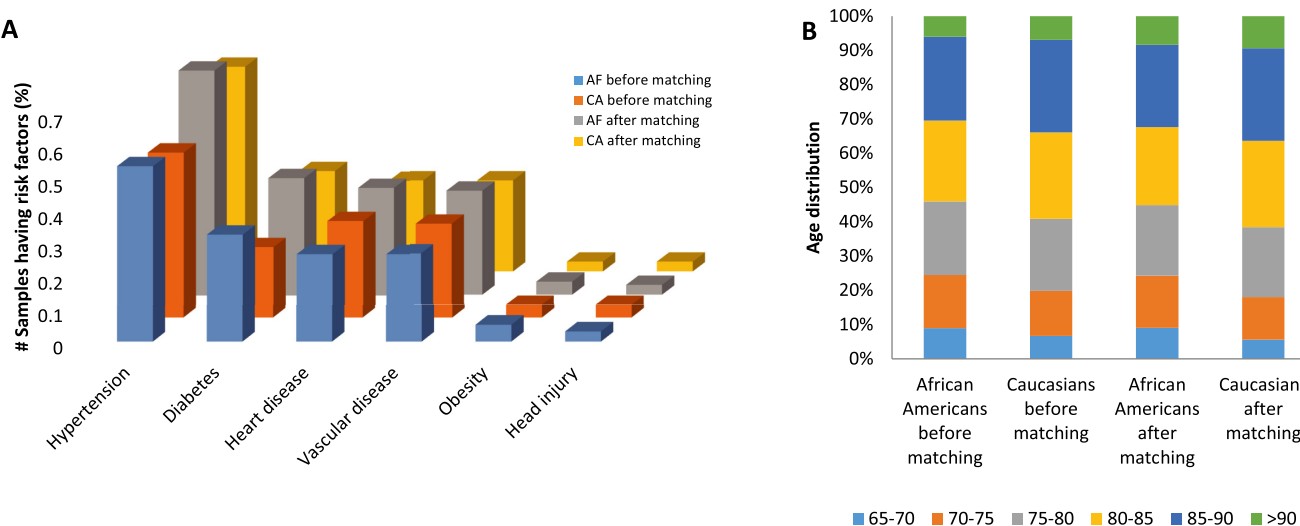

**Fig 2. Subject's age and known risk factor distributions.** (a) Onset age distribution of ADRD patients after matching Caucasians to African Americans. The onset age of (original) Caucasians tends to be older than that of African Americans. The matched Caucasian follows a similar distribution. (b) Distribution of known risk factors after matching. The matched Caucasian follows a similar distribution with African Americans in terms of the confounding risk factors. Ca = Caucasians, Paired Ca = Caucasians that are matched to African Americans. AF = African Americans.

implies that African Americans with *the late effect of CVD* would have decreased the risk for ADRD by 19.08% point if their CVD was treated. We confirmed the ATE estimate via the 100 repeated permutation tests [62]. The estimated ATE of the permuted variable was -0.0062 (*p*-value = 0.34), which means rejection of Fisher's Sharp Null; therefore, the treatment effect is present. We examined the Bayesian network of the late effects of CVD and highlighted a few paths to the late effects of CVD and ADRD (Fig 3A). In addition, the African American's Bayesian network had an edge from *transient cerebral ischemia* (PheWas Code = *433.31*, specific diseases it covers are in S3 Table) to ADRD, with the number of appearances of seven (S2 Table). Transient cerebral ischemia had ATE of 0.1421 and ATT of 0.0242, implying a similar treatment effect to the late effects of CVD.

**Mental disorders.** Both racial groups had multiple edges from mental disorders to ADRD: *presenile dementia* (290.1), *other persistent mental disorders* (290.3), *memory loss* (292.3), *altered mental status* (292.4), and *Parkinson's disease* (332). These disorders might represent prodromal status (e.g., mild cognitive impairment) before ADRD onset.

The Caucasian counterparts had a direct edge from *depression* (296.2) to ADRD, whereas African Americans did not. The ATE (IPTW) of *depression* on the Caucasian counterparts was ATE = 0.1557 (Table 4), which implies that having *depression* increases the ADRD risk by 15.60% point on average. Similarly, the ATT of 0.1154 implies that Caucasian counterparts with depression would have decreased the risk for ADRD by 11.92% point if their *depression* was treated. During the ATE confirmation via the permutation test, we found -0.0005 (*p*-value = 0.48) for the permuted variables. We examined the Bayesian network of depression and highlighted a few paths to depression and ADRD (Fig 3B).

**Inflammation/Infection.** African Americans had edges from *urinary tract infection* (591) and *acute upper respiratory infections* (465) to ADRD. For the urinary tract infection, ATE = 0.1149 with 95% confidence interval (CI) = (0.0918, 0.1371) means urinary tract infection increases the ADRD risk by 11.49% point on average; ATT = 0.0573 with CI = (0.0343, 0.0903) means treating urinary tract infection decreases the ADRD risk by 5.73% points. For acute upper respiratory infections, ATE = -0.2554 with CI = (-0.2984, -0.2373) means acute

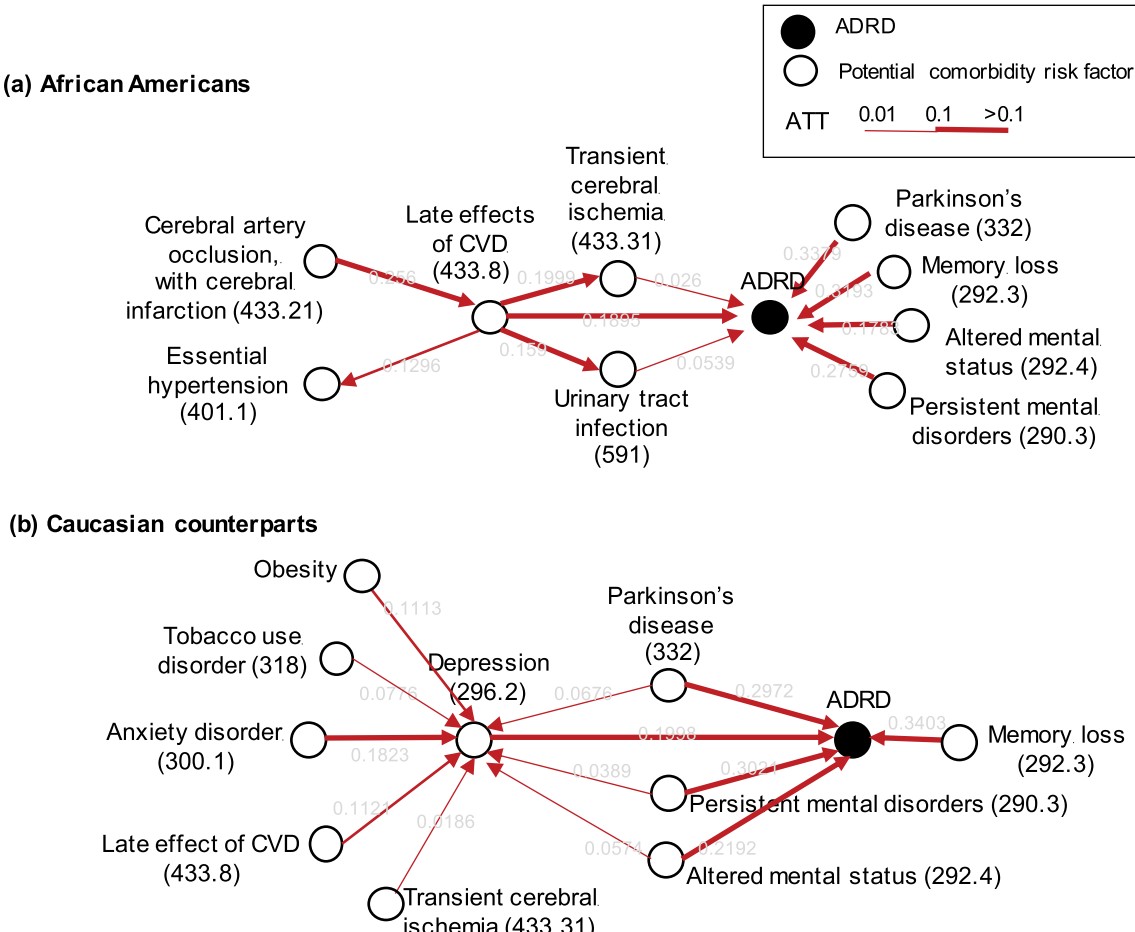

**Fig 3.** Bayesian Networks developed from the Cerner EHR Dataset that illustrates the differential risks for ADRD in African Americans (a) and Caucasians (b). We highlighted the major identified variables that contribute to the risk for developing the late effects of CVD and how CVD contributes to the risk for ADRD in older African Americans (a). Cerebral artery occlusion and cerebral infarction lead to late effects of CVD, and the late effect of CVD leads to transient cerebral ischemia or ADRD directly (a). The full Bayesian network is in S2 Table.

**Table 3. Comorbidities (PheWas disease code) with a direct edge to ADRD in Bayesian Network.** We selected edges that appeared more than nine times out of ten repetitions. More edges in Bayesian Network are at S2 Table.

| Comorbidity type | Comorbidities that have directed edge to ADRD | | |
|---|---|---|---|
| | **African Americans** | **Both** | **Caucasian counterparts** |
| Cerebrovascular disease (CVD) | **Late effects of CVD (433.8)** | - | - |
| Mental disorders | - | Senile dementia (290.1), Other persistent mental disorders (290.3), Memory loss (292.3), Altered mental status (292.4), Parkinson's disease (332). | **Depression (296.2)** |
| Inflammation/Infection | Acute upper respiratory infections (465), Urinary tract infection (591) | - | - |

**Table 4. Treatment effect of comorbidity on ADRD onset (95% confidence interval).** ATE = Average treatment effect, ATT = Average treatment effect among treated (ATE among the subjects with the comorbidity), ATC = Average treatment effect among controls (ATE among the subjects without the comorbidity). IPTW = Inverse probability of treatment weighting.

| Unique comorbidity in each racial group | Type of treatment effect | Algorithms to estimate the average treatment effect | | |
|---|---|---|---|---|
| | | Propensity score matching | Propensity score stratification | IPTW |
| Treatment effect of *the late effect of cerebrovascular disease* on ADRD onset in older African Americans | ATE | 0.3121 (0.2210, 0.3486) | 0.0802 (0.0572, 0.1537) | 0.2715 (0.2423, 0.3137) |
| | ATT | 0.1107 (0.0466, 0.1736) | 0.0486 (0.0291, 0.1180) | 0.1908 (0.1554, 0.2350) |
| | ATC | 0.3216 (0.2158, 0.3597) | 0.0855 (0.0690, 0.1767) | 0.27865 (0.2467, 0.3133) |
| Treatment effect of *depression* on ADRD onset in Caucasian counterparts | ATE | 0.0725 (0.0753, 0.1765) | 0.1207 (0.0696, 0.1506) | 0.1557 (0.1293, 0.1852) |
| | ATT | 0.0796 (0.0538, 0.1666) | 0.0924 (0.0617, 0.11129) | 0.1154 (0.0716, 0.1507) |
| | ATC | 0.0720 (0.0636, 0.2021) | 0.1249 (0.0699, 0.1517) | 0.1600 (0.1294, 0.1864) |

upper respiratory infections decrease the ADRD risk by 25.54% point on average; ATT = -0.2662 with CI = (-0.2827,-0.2327) means treating acute upper respiratory infections increase the ADRD risk by 26.62% points. Further investigation is needed for the mixed effects of inflammation, such as considering the confounding effect of inflammation itself and/or medications related to inflammation. For example, Zileuton, a leukotriene biosynthesis inhibitor that is widely used for chronic inflammation (asthma), has shown a significantly reduced level of neuroinflammation and in tau phosphorylation in the tau transgenic mice [65].

## Discussion and conclusions

The objective of this study was to investigate differential effects of comorbidities that increase the risk of ADRD in African Americans compared to non-Hispanic Caucasians using counterfactual analysis of a nationwide EHR. We matched African Americans and Caucasians based on age, sex, and high-risk common comorbidities. Using the learned Bayesian network, average treatment effect estimation, and permutation test confirmation, we identified that the *late effects of CVD* predispose older African Americans to ADRD, but we did not find the same strength of this counterfactual effect in Caucasians. Our counterfactual approach to identify ADRD risk factors has methodological innovation in that we first address causation (not association) in comorbidities by integrating an ensemble causal structure model and counterfactual treatment effect estimation to better understand the root cause of ADRD risk. Our findings have clinical implications in that the identified ADRD risk factors (*late effects of CVD*) and the factors leading towards the later stages of CVD could potentially be modifiable.

In our study, we found that there are several significant differences between the comorbidities contributing to the development of ADRD among the two racial groups. African Americans had statistically significant counterfactual effects from CVD on ADRD risk, which implies that the late effects of CVD in African Americans are more likely to predispose patients to ADRD than in Caucasians. CVD in African Americans may lead to vascular dementia, Alzheimer's, or both to a greater extent than Caucasians. For the same disease burden, CVD may directly lead to the advancement of the pathology intrinsic to ADRD or subsequent transient ischemic attack (TIA) may accelerate the disease process towards ADRD. It is known that African Americans have higher incidences of CVD, small vessel disease, and vascular risk factors than Caucasians but even when controlling for vascular risk factors, African Americans have a

higher risk for vascular end organ complications compared with Caucasians. TIAs by themselves are manifestations of uncontrolled cerebrovascular disease such as atherosclerosis of the extracranial and intracranial arterial circulation and may indicate harbingers of advancing disease towards vascular dementia, Alzheimer's, or both. We considered other possibilities. African Americans could be under-treated with antiplatelets and statins leading to more TIAs and advancement towards ADRD. Although we do not yet understand the mechanisms of how TIAs would lead to ADRD, these findings suggest that TIAs may be a controllable risk factor that can reduce the risk of ADRD onset in African Americans. Furthermore, the late effects of CVD could still be associated with ADRD in Caucasians. Our findings indicate that the late effects of CVD do not have a counterfactual treatment effect in Caucasians.

In our analysis of matched Caucasian counterparts, we found a very different set of comorbidities pathways leading to ADRD. Depression was one of the major factors that contributed to ADRD [66–68]. We identified several pathways leading to depression in the EHR dataset. For Caucasians, depression and ADRD are commonly observed together and share many symptoms in older adults. Depression has been associated with poor cognitive functions [69] and increases the risk of ADRD [67,70,71]. The actual causal relationships between depression and ADRD are still controversial [67,72]. For African Americans it has been long known that there are racial disparities in depression and quality of care [73]. In general, African Americans have poorer mental health than non-Hispanic Caucasians [74]. Nonetheless, depressed African Americans are less likely to utilize mental health care than non-Hispanic Caucasians [73,75,76], due to socio-cultural factors including racism, misdiagnosis, and clinician bias [77]. We believe that the depression which is only observed in Caucasian counterparts in our Bayesian network analysis confirms racial disparities in seeking mental care. Therefore, we acknowledge that depression could be a more significant risk factor for ADRD than we were able to detect.

There are several limitations of this study. The data in EHR are observations that cannot fully represent the entire population, for example, the population in the Cerner EHR is mostly from the middle class that is privately insured. Another limitation is that ADRD is a progressive disease that slowly develops over several years to decades. EHRs might not be able to fully capture such a long-life course progression. The median observation period was 1.0 years (min was 0.5 years, max 11 years), implying the non-ADRD subjects might be censored too early to observe their ADRD risk. Late ADRD onset age (due to social determinants) also shows the limitation of EHRs. In addition, there have been long debates whether biological characteristics such as race or comorbidities should be evaluated as causes even if intervention to such variables is difficult [24]. From the methodology perspective, the cohort matching method can potentially cause bias as the unmatched patients were excluded in the analysis. Quantitative evaluation of the Bayesian network was not available due to the lack of ground truth. A potential way to build a better causal structure is to incorporate prior knowledge mined from literature. We obtained the causal structure in a data-driven way without incorporating the literature-based prior knowledge. Injecting such prior knowledge into the causal structure might allow us to seamlessly harmonize existing knowledge and new data-driven causality. Notwithstanding these limitations, this study offers some important insights into racially differential comorbidity risk factors in ADRD. In addition, despite the noisy and incomplete nature of the real-world data, the counterfactual analysis on the comorbidity risk factors can be a valuable tool to support the risk factor exposure studies.

## Supporting information

**S1 Text.** A. Definition of ADRD in EHRs, B. Cohort matching methods, C. Disentangle the dependency among comorbidities and ADRD.
(PDF)

**S1 Table.** Selected known comorbidities that we controlled to build a balanced cohort.
(DOCX)

**S2 Table.** Causal structure of African Americans and Caucasian counterparts.
(XLSX)

**S3 Table.** Definition of late effect of cerebrovascular disease and transient cerebral ischemia.
(DOCX)

## Author Contributions

**Conceptualization:** Yejin Kim, Sean I. Savitz, Paul E. Schulz, Xiaoqian Jiang.

**Data curation:** Yejin Kim, Luyao Chen, Xiaoqian Jiang.

**Formal analysis:** Yejin Kim, Kai Zhang, Sean I. Savitz, Xiaoqian Jiang.

**Funding acquisition:** Xiaoqian Jiang.

**Investigation:** Yejin Kim, Kai Zhang, Sean I. Savitz, Luyao Chen, Paul E. Schulz, Xiaoqian Jiang.

**Methodology:** Yejin Kim, Kai Zhang, Xiaoqian Jiang.

**Resources:** Xiaoqian Jiang.

**Software:** Yejin Kim, Kai Zhang, Luyao Chen.

**Supervision:** Sean I. Savitz, Paul E. Schulz, Xiaoqian Jiang.

**Validation:** Sean I. Savitz, Paul E. Schulz, Xiaoqian Jiang.

**Visualization:** Yejin Kim.

**Writing – original draft:** Yejin Kim, Kai Zhang, Sean I. Savitz, Luyao Chen, Paul E. Schulz, Xiaoqian Jiang.

**Writing – review & editing:** Yejin Kim, Kai Zhang, Sean I. Savitz, Paul E. Schulz, Xiaoqian Jiang.

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
