## [Decision Letter · Decision Letter 0]

1 Sep 2021

PDIG-D-21-00037

Counterfactual Analysis of Differential Comorbidity Risk Factors in Alzheimer’s Disease and Related Dementias

PLOS Digital Health

Dear Dr. Kim,

Thank you for submitting your manuscript to PLOS Digital Health. After careful consideration, we feel that it has merit but does not fully meet PLOS Digital Health’s publication criteria as it currently stands. Therefore, we invite you to submit a revised version of the manuscript that addresses the points raised during the review process.

We look forward to receiving your revised manuscript.

Kind regards,

Dukyong Yoon

Academic Editor

PLOS Digital Health

Journal Requirements:

1. We ask that a manuscript source file is provided at Revision. Please upload your manuscript file as a .doc, .docx, .rtf or .tex. If you are providing a .tex file, please upload it under the item type ‘LaTeX Source File’ and leave your .pdf version as the item type ‘Manuscript’.

2. Please provide separate figure files in .tif or .eps format only, and remove any figures embedded in your manuscript file.  If you are using LaTeX, you do not need to remove embedded figures.

For more information about figure files please see our guidelines: https://journals.plos.org/digitalhealth/s/figures

3. We do not publish any copyright or trademark symbols that usually accompany proprietary names, eg (R), (C), or TM  (e.g. next to drug or reagent names). Therefore please remove all instances of trademark/copyright symbols throughout the text, including Cerner Health Facts ® on page 29.

4. We have noticed that you have uploaded supporting information but you have not included a list of legends.  Please add a full list of legends for all supporting information files (including figures, table and data files) after the references list.

Additional Editor Comments (if provided):

The provided data by the authors (Supplementary Material 2 and 3) is not enough to replicate all study findings reported in the article.

Please see our policy on "Data Availability" at following link.

https://journals.plos.org/digitalhealth/s/data-availability

Reviewers' comments:

Reviewer's Responses to Questions

**Comments to the Author**

1. Does this manuscript meet PLOS Digital Health’s publication criteria? Is the manuscript technically sound, and do the data support the conclusions? The manuscript must describe methodologically and ethically rigorous research with conclusions that are appropriately drawn based on the data presented.

Reviewer #1: Yes

Reviewer #2: Yes

Reviewer #3: Yes

2. Has the statistical analysis been performed appropriately and rigorously?

Reviewer #1: Yes

Reviewer #2: I don't know

Reviewer #3: Yes

3. Have the authors made all data underlying the findings in their manuscript fully available (please refer to the Data Availability Statement at the start of the manuscript PDF file)?

Reviewer #1: Yes

Reviewer #2: No

Reviewer #3: Yes

4. Is the manuscript presented in an intelligible fashion and written in standard English?

Reviewer #1: Yes

Reviewer #2: Yes

Reviewer #3: Yes

5. Review Comments to the Author

Reviewer #1: Comments to the Author:

Thank you for letting me review the paper “Counterfactual Analysis of Differential Comorbidity Risk Factors in Alzheimer’s Disease and Related Dementia”. The study investigated counterfactual treatment effects of various comorbidity in Alzheimer’s disease and related dementia. The study applied very interesting methodology and yields very interesting results. However, I have some concerns that applies to the methods and results

1. I wonder what the difference is between counterfactual analysis and adding the propensity score as a variable in the regression model.

2. I wonder if there are any similar health care studies or examples using Counterfactual effect with Bayesian networks.

3. I am curious how to interpret the result of Average Treatment Effect(ATE). In the field of public health or medicine, it is important to identify the types of disease causes and measure the size of risk (i.e., odds ratio or hazard ratio). In particular, the interpretation of risk is very important. For example, smokers are four to five times more likely to develop lung cancer than non-smokers. Please describe how it is desirable to interpret ATE.

Reviewer #2: This paper tried to find the unknown comorbidities that can cause ADRD through the Causal Bayesian network. In particular, differences in ADRD mechanisms between races were confirmed by constructing different networks for each race: African Americans and Caucasians.

I would like to give two major opinions and two minor opinions.

#1 Major opinion

In this study, a Causal Bayesian network was used to find unknown ADRD-inducing comorbidities.

However, this study does not provide quantitative results that the proposed method is effective in finding ADRD-inducing comorbidities. To prove that the proposed model can detect ADRD-inducing comorbidities, it should be suggested to show how many of the known comorbidities are detected through the proposed model when the proposed model is analyzed without matching the known comorbidities. Then, reliability can be obtained even for the result of finding the unknown comorbidities.

#2 Major opinion

The constructed Bayesian network itself could be a biased network for this study data. please show whether building a network multiple times with different subsamples has the similar result. Moreover, I hope you can prove that the networks between the two races are significantly different.

#3 Minor opinion

Please provide the p-value whether the matched result statistically removed the difference significantly in Table 1.

#4 Minor opinion

In Supplementary S2, please write ADRD's Markov blanket by name and code together for readability.

Reviewer #3: In this paper, the authors compare counterfactual treatment effects of various comorbidity in ADRD in different racial groups. They derived a Bayesian network of 100 comorbidities and selected comorbidities with potential causal effect to ADRD. They then estimated the ATE of the selected comorbidities on ADRD using IPTW. While this is an important question, there are some concerns about the framing of the study and the approach taken.

1. Overall, there are several strengths to this study. The goal of the study is very timely, and the methods used are promising. In addition, authors utilized a large clinical database covering EHRs from more than 600 Cerner client hospitals which makes the results more interesting.

2. The first major consideration is about the Bayesian network. To disentangle the multifactorial effect of comorbidities, the authors derived a Bayesian network of comorbidities and ADRD using constraint-based algorithms. Then the authors estimated the ATE of the selected comorbidities on ADRD. How to evaluate the results of the Bayesian network? Because a different method may result in different selected comorbidities.

3. If the authors estimated the ATE of all the comorbidities, what will happen? Is it possible that some of the unselected comorbidities (by Bayesian network) may have higher/lower average treatment effect?

4. The key elements of study design are unclear, e.g., observation period, following up period, index date. Please add further details and clarification.

5. How does the study design deal with the censored patients?

6. The authors used inverse probability of treatment weighting to estimate the counterfactual effect of comorbidities that have a direct edge to ADRD in each racial group’s Bayesian network. To replicate the study and demonstrate the randomized emulation, the setting and results of each step of IPTW should be provided, e.g., unbalanced ratio, SMD threshold.

6. PLOS authors have the option to publish the peer review history of their article (what does this mean?). If published, this will include your full peer review and any attached files.

**Do you want your identity to be public for this peer review?** For information about this choice, including consent withdrawal, please see our Privacy Policy.

Reviewer #1: No

Reviewer #2: No

Reviewer #3: No

---

## [Decision Letter · Decision Letter 1]

5 Dec 2021

PDIG-D-21-00037R1

Counterfactual Analysis of Differential Comorbidity Risk Factors in Alzheimer’s Disease and Related Dementias

PLOS Digital Health

Dear Dr. Kim,

Thank you for submitting your manuscript to PLOS Digital Health. After careful consideration, we feel that it has merit but does not fully meet PLOS Digital Health’s publication criteria as it currently stands. Therefore, we invite you to submit a revised version of the manuscript that addresses the points raised during the review process.

We look forward to receiving your revised manuscript.

Kind regards,

Dukyong Yoon

Academic Editor

PLOS Digital Health

Additional Editor Comments (if provided):

Reviewers' comments:

Reviewer's Responses to Questions

**Comments to the Author**

1. If the authors have adequately addressed your comments raised in a previous round of review and you feel that this manuscript is now acceptable for publication, you may indicate that here to bypass the “Comments to the Author” section, enter your conflict of interest statement in the “Confidential to Editor” section, and submit your "Accept" recommendation.

Reviewer #1: All comments have been addressed

Reviewer #2: (No Response)

2. Does this manuscript meet PLOS Digital Health’s publication criteria? Is the manuscript technically sound, and do the data support the conclusions? The manuscript must describe methodologically and ethically rigorous research with conclusions that are appropriately drawn based on the data presented.

Reviewer #1: Yes

Reviewer #2: Yes

3. Has the statistical analysis been performed appropriately and rigorously?

Reviewer #1: Yes

Reviewer #2: No

4. Have the authors made all data underlying the findings in their manuscript fully available (please refer to the Data Availability Statement at the start of the manuscript PDF file)?

Reviewer #1: Yes

Reviewer #2: Yes

5. Is the manuscript presented in an intelligible fashion and written in standard English?

Reviewer #1: Yes

Reviewer #2: Yes

6. Review Comments to the Author

Reviewer #1: (No Response)

Reviewer #2: The outcomes that have been provided are still subjective. We expect that definitions, outcomes, and interpretations are all provided clearly when presenting results.

A. In Supplementary 2, how many times must it be detected among the ADRSs to be regarded a meaningful outcome? It is essential to define the criteria or explain the rationale advance.

B. In Supplementary 4, A Venn diagram was used to show the differences in node across racial groups. However, a threshold for declaring difference between the two groups must be established in advance.

7. PLOS authors have the option to publish the peer review history of their article (what does this mean?). If published, this will include your full peer review and any attached files.

**Do you want your identity to be public for this peer review?** For information about this choice, including consent withdrawal, please see our Privacy Policy.

Reviewer #1: No

Reviewer #2: No

---

## [Decision Letter · Decision Letter 2]

21 Jan 2022

Counterfactual Analysis of Differential Comorbidity Risk Factors in Alzheimer’s Disease and Related Dementias

PDIG-D-21-00037R2

Dear Dr Kim,

We are pleased to inform you that your manuscript 'Counterfactual Analysis of Differential Comorbidity Risk Factors in Alzheimer’s Disease and Related Dementias' has been provisionally accepted for publication in PLOS Digital Health.

Best regards,

Dukyong Yoon

Academic Editor

PLOS Digital Health

Reviewer Comments (if any, and for reference):

Reviewer's Responses to Questions

**Comments to the Author**

1. If the authors have adequately addressed your comments raised in a previous round of review and you feel that this manuscript is now acceptable for publication, you may indicate that here to bypass the “Comments to the Author” section, enter your conflict of interest statement in the “Confidential to Editor” section, and submit your "Accept" recommendation.

Reviewer #2: All comments have been addressed

2. Does this manuscript meet PLOS Digital Health’s publication criteria? Is the manuscript technically sound, and do the data support the conclusions? The manuscript must describe methodologically and ethically rigorous research with conclusions that are appropriately drawn based on the data presented.

Reviewer #2: Yes

3. Has the statistical analysis been performed appropriately and rigorously?

Reviewer #2: Yes

4. Have the authors made all data underlying the findings in their manuscript fully available (please refer to the Data Availability Statement at the start of the manuscript PDF file)?

Reviewer #2: Yes

5. Is the manuscript presented in an intelligible fashion and written in standard English?

Reviewer #2: Yes

6. Review Comments to the Author

Reviewer #2: All comments have been addressed

7. PLOS authors have the option to publish the peer review history of their article (what does this mean?). If published, this will include your full peer review and any attached files.

**Do you want your identity to be public for this peer review?** For information about this choice, including consent withdrawal, please see our Privacy Policy.

Reviewer #2: No
